# Multi-Label Learning with Contrastive Cluster Self-Supervision for 3D Hierarchical Semantic Segmentation

Shuyu Cao [‡ 1]   Chongshou Li [‡ 2]   Jie Xu[* 3]   Tianrui Li [2]   Na Zhao [3]

## Abstract

3D hierarchical semantic segmentation (3DHS) is crucial for embodied intelligence that demands the coarse-to-fine grained and multi-hierarchy understanding of 3D scenes. 3DHS tasks can be addressed by multi-label learning, but facing two issues: I) learning multiple labels for each point with a shared model can lead to multi-hierarchy conflicts in cross-hierarchy optimization, and II) the class imbalance issue is inevitable across multiple hierarchies of 3D scenes, making the model easily be dominated by major classes. To address these issues, we propose a novel multi-label learning with contrastive cluster self-supervision framework for 3DHS. Specifically, we propose a late-decoupled multi-label learning 3DHS network which employs decoupled decoders with the coarse-to-fine hierarchical consistency guidance. This late-decoupled model architecture can mitigate the underfitting and overfitting conflicts among multiple hierarchies and also constrain the class imbalance problem within each individual hierarchy. Moreover, we introduce a 3DHS-oriented contrastive cluster self-supervision learning method, which learns cluster-wise point cloud features with contrastive loss and produces self-supervised information to enhance the class-imbalance segmentation. Extensive experiments on multiple datasets and backbones demonstrate that our approach promotes the multi-hierarchy balance and mitigates the class imbalance issue in 3DHS tasks.

## 1. Introduction

3D semantic segmentation (3DS) is a fundamental task in 3D vision, aiming to assign a single semantic label to every point in a 3D scene (Milioto et al., 2019; Zhao et al., 2021b; Li & Zhao, 2024). Pioneering 3DS methods leverage point cloud oriented backbone networks such as PointNet (Qi et al., 2017a), PointNet++ (Qi et al., 2017b), Point Transformer (Zhao et al., 2021a; Guo et al., 2021; Wu et al., 2024a) to extract point cloud features and use a classifier to learn point-wise labels. In practice, however, an object usually possesses rich semantic meanings that go beyond a single label, e.g., a table belongs to both the furniture category and the wood product category, making traditional single-hierarchy 3DS models unable to satisfy the requirements in real-world applications. This detailed, hierarchy-aware scene understanding is crucial for a wide range of future applications, from enabling rich interactive experiences in 3D space in augmented reality (Dai et al., 2017; McCormac et al., 2017; Wu et al., 2023; 2024b) to creating safe navigation in autonomous driving and automatic manipulation in embodied robotics (Behley et al., 2019; Pan et al., 2026; Li et al., 2024b; Feng et al., 2025). To this end, the field has evolved towards 3D hierarchical semantic segmentation (3DHS) aiming at learning multi-hierarchy semantic labels for every point, and recently, 3DHS tasks have garnered significant attention from the research community (Li et al., 2020; 2025).

To be specific, due to the single-hierarchy design, previous 3DS models are architecturally ill-suited for 3DHS tasks. Moreover, training an individual 3DS model for each hierarchy is costly and there is a lack of positive interaction among multiple hierarchies (Yu et al., 2019; Mo et al., 2019; Roberts et al., 2021). Therefore, researchers have recently devoted their efforts to innovating in the hierarchical model structure and cross-hierarchy interaction for 3DHS. For example, pioneering work designed a multi-task hierarchical segmentation network MTHS (Li et al., 2020) that stacks multiple parallel classifiers on a shared point cloud encoder-decoder and leverages a multi-task classification loss to achieve 3DHS. Furthermore, Li et al. (2025) proposed a model-agnostic deep hierarchical learning (DHL) method, which adopts the top-bottom and bottom-top fu-

‡Equal contribution *Corresponding author [1]SWJTU-Leeds Joint School, Southwest Jiaotong University, Chengdu, China [2]School of Computing and AI, Southwest Jiaotong University, Chengdu, China [3]Singapore University of Technology and Design, Singapore. Correspondence to: Jie Xu <jiexu-work[AT]outlook[DOT]com>.

*Proceedings of the 43$^{rd}$ International Conference on Machine Learning*, Seoul, South Korea. PMLR 306, 2026. Copyright 2026 by the author(s).

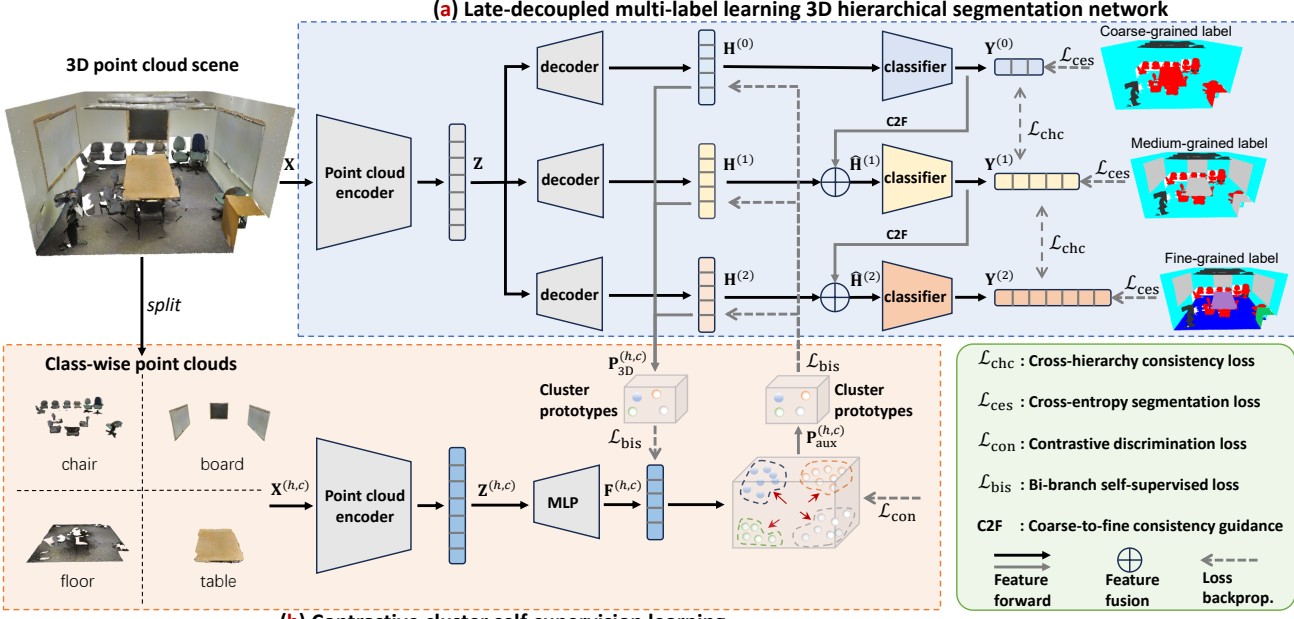

*Figure 1.* Overview of ML3DHS. (a) The late-decoupled multi-label learning 3DHS network uses a shared point cloud encoder with multiple late-decoupled decoders to perform hierarchical segmentation tasks, and leverages a coarse-to-fine guidance mechanism to balance task-specific learning and hierarchical consistency. Meanwhile, (b) the 3DHS-oriented contrastive cluster self-supervision learning method employs contrastive learning to learn discriminative representations for individual classes, and then provides bi-directional self-supervision with the above 3DHS network, thereby improving the segmentation ability on class-imbalance point clouds.

sion mechanisms and optimizes the cross-hierarchy consistency loss to pass information across hierarchies.

Although existing 3DHS methods achieved important progress, they still face two challenges. I) *Multi-hierarchy conflicts*: To achieve 3DHS, we usually need to learn multiple labels for different hierarchies with the shared parameters based on 3DS models (Qi et al., 2017b;a; Zhao et al., 2021a; Li et al., 2025). This often requires manually designed balancing strategies for different hierarchies' label learning (e.g., distinct loss weights for hierarchical optimization objectives). However, there are conflicts among hierarchies like the model overfitting one and underfitting the other, which makes balancing difficult and limits the entire model's effectiveness (Wen et al., 2026). II) *Class imbalance issue*: The point class imbalance is inevitable in 3D segmentation domains (Li et al., 2020) especially for 3DHS where the number of hierarchies increases, resulting in the model being easily dominated by majority classes (e.g., wall, floor) and the segmentation performance on minority classes (e.g., column, board) might degenerate.

To address these challenges, we propose *multi-label learning with contrastive cluster self-supervision* for 3DHS. As shown in Figure 1, our framework entitled ML3DHS is driven by two insights. *First*, we formally consider the multi-hierarchy segmentation task to be analogous to a multi-label learning task (Sener & Koltun, 2018; Chen

et al., 2018; Yu et al., 2020) which suggests using the shared parameters to fit different hierarchies' labels will lead to the overfitting at some hierarchies and underfitting at others. This creates the multi-hierarchy conflicts among the optimization objectives of different hierarchies in our context. To mitigate this issue, we propose to establish a late-decoupled framework for each hierarchical segmentation task, while multiple hierarchies share one point cloud feature extraction encoder to ensure a consistent information foundation. *Second*, the class imbalance drawback is expected to be constrained in each individual hierarchy due to the late-decoupled architecture. Furthermore, motivated by contrastive learning (Chen et al., 2020), we incorporate hierarchy-specific contrastive cluster self-supervision, which structures the latent space into discriminative clusters and encourages the model to capture cluster-wise discriminative information. This mechanism effectively mitigates the optimization bias inherent in class-imbalanced hierarchical learning.

In summary, the contributions of this work are threefold:

- We propose a theoretically-grounded, late-decoupled multi-label learning 3DHS network which employs multiple decoders with the coarse-to-fine hierarchical guidance and consistency, to alleviate the conflicts arising from multi-label learning and promote its application in 3D hierarchical segmentation tasks.

- We introduce a flexible 3DHS-oriented contrastive cluster self-supervision learning method, which learns cluster-wise point cloud features with contrastive loss and produces self-supervised discriminative information to enhance the class-imbalance segmentation.

- Our approach achieves state-of-the-art 3DHS performance across multiple 3D benchmark datasets and backbones, and can be integrated into previous methods to further improve their 3DHS performance.

## 2. Related Work

### 2.1. Deep 3D semantic segmentation

Recently, 3D semantic segmentation (3DS) has achieved significant progress with the development of deep models. For example, PointNet (Qi et al., 2017a) is a pioneering work that directly learns point features with multilayer perceptron instead of using handcrafted features and is widely employed to achieve deep 3DS. Furthermore, PointCNN (Xu et al., 2018) leverages 3D convolutional networks to improve point cloud feature learning. Motivated by the success of deep 3DS, many efforts have been made in this community and proposed a series of gradually developed methods, such as KPConv (Thomas et al., 2019; 2024), MinkowskiNet (Choy et al., 2019), PVCNN (Liu et al., 2019), DGCNN (Wang et al., 2019), and Point Transformer (Guo et al., 2021; Zhao et al., 2021a). Some work (Milioto et al., 2019; Zhou et al., 2020; Ma et al., 2022) focuses on developing efficient methods to process large-scale point cloud scenes. These advances are supported by large-scale datasets such as S3DIS (Armeni et al., 2016), Campus3D (Li et al., 2020) and SensatUrban (Hu et al., 2022). However, existing 3DS methods usually ignore the hierarchical relationships in real-world scenes (Li et al., 2020; Zhao et al., 2023; Pan et al., 2025; Yuan et al., 2026). They can only predict flat semantic labels and might suffer from performance degradation under class imbalance (Peri et al., 2022; Li et al., 2024a). In this paper, we are based on the previously established point cloud network to propose a hierarchy-aware deep 3DS framework while mitigating the negative effect of class imbalance.

### 2.2. 3D hierarchical semantic segmentation

The concept of hierarchical segmentation, first explored in 2D image analysis for multi-hierarchy understanding, has recently been adapted to 3D point clouds. Specifically, hierarchical semantic segmentation approaches (Farabet et al., 2013; Lin et al., 2017; Zhao et al., 2017) show that combining coarse vision features with fine ones can substantially improve segmentation accuracy. This observation highlights the importance of modeling semantic structures across multiple hierarchies for establishing 3D hierarchical semantic segmentation (3DHS). To this end,

Campus3D (Li et al., 2020), a large-scale outdoor point cloud benchmark with multi-hierarchy annotations, was introduced to support training 3DHS models. Then, Li et al. (2025) proposed deep hierarchical learning to promote cross-hierarchy semantic consistency in 3DHS. Although these 3DHS methods achieved progress, they still face the class imbalance challenge. Moreover, previous work has ignored the conflicts among multi-hierarchy labels when aligning deep models (Xu et al., 2024; 2025). In this paper, we propose a novel late-decoupled multi-label learning 3DHS framework to address these issues.

### 2.3. Self-supervised learning

To reduce reliance on costly annotations, self-supervised learning (SSL) has been widely explored for 3D vision. For instance, PointContrast (Xie et al., 2020) learns robust features by optimizing contrastive loss to align point features of different views. Pang et al. (2022) reconstruct masked point cloud patches to train 3D representation learning models. SSL-based 3D segmentation methods (Zhang et al., 2023; Yu et al., 2022) show promising directions for label-efficient segmentation via superpoint growth and clustering, but they lack explicit hierarchical supervision and struggle to guarantee cross-hierarchy semantic coherence. Recent work has started to address class imbalance issues by using self-supervised methods (Han et al., 2024; 2025) and semi-supervised frameworks (Li et al., 2024a). These methods demonstrate that SSL guidance is effective for improving rare class performance. Motivated by the success of SSL (Xu et al., 2023; Chen et al., 2025; Zhao et al., 2025; Chen et al., 2026), in this work, we leverage contrastive learning and construct a self-supervision mechanism to improve model discrimination ability for addressing the class imbalance in 3DHS tasks.

## 3. Method

### 3.1. Problem definition

The 3D hierarchical semantic segmentation (3DHS) task is to predict hierarchical semantic labels for each point in a 3D point cloud scene. Formally, given a point cloud $\mathbf{X} \in \mathbb{R}^{N \times 3}$, 3DHS methods learn a function $\mathcal{F}$ that predicts a set of labels $\{l_i^{(1)}, l_i^{(2)}, \ldots, l_i^{(H)}\}$ with hierarchical structures for each point $\mathbf{x}_i \in \mathbf{X}$, where $N$ denotes the point number and $H$ is the hierarchy number. Concretely, the 3DHS model can be formulated as follows:

$$\{\mathbf{Y}^{(h)}\}_{h=1}^H = \mathcal{F}_\theta(\mathbf{X}), \tag{1}$$

where $\mathbf{Y}^{(h)} = [\mathbf{y}_1^{(h)}; \mathbf{y}_2^{(h)}; \ldots; \mathbf{y}_N^{(h)}] \in \mathbb{R}^{N \times K^{(h)}}$, $\mathbf{y}_i^{(h)} \in \mathbb{R}^{K^{(h)}}$ represents the probabilistic classification results of the $i$-th point in the $h$-th hierarchy, $K^{(h)}$ is the class number. $\theta$ are the model parameters to be trained. As a result,

$l_i^{(h)} = \arg\max_j y_{i,j}^{(h)}$, $y_{i,j}^{(h)} \in \mathbf{y}_i^{(h)}$. To address the multi-hierarchy conflict and class imbalance issue, we propose a novel framework ML3DHS as shown in Figure 1, which establishes a *late-decoupled multi-label learning 3DHS network* to avoid multi-hierarchy conflicts as well as trains a *3DHS-oriented contrastive cluster self-supervision learning* branch for mitigating class imbalance issue. The following sections introduce the method details.

### 3.2. Late-decoupled multi-label learning with coarse-to-fine hierarchical consistency guidance

**Late-decoupled multi-label learning for 3DHS.** We are motivated by the insight that shared parameters might cause multi-view optimization conflicts (Xu et al., 2024). In Figure 1(a), we propose the late-decoupled 3DHS branch to avoid multi-hierarchy conflicts. Specifically, our method improves the conventional formulation of Eq. (1) as:

$$\{\mathbf{Y}^{(h)}\}_{h=1}^H = \{\mathcal{G}_{\delta^{(h)}}^{(h)}(\mathbf{Z})\}_{h=1}^H = \{\mathcal{G}_{\delta^{(h)}}^{(h)}(\mathcal{E}_\theta(\mathbf{X}))\}_{h=1}^H, \quad (2)$$

where $\mathbf{Z} = \mathcal{E}_\theta(\mathbf{X})$ denotes the early point cloud feature extraction, while $\mathbf{Y}^{(h)} = \mathcal{G}_{\delta^{(h)}}^{(h)}(\mathbf{Z})$ indicates that we establish the late-decoupled 3DHS branch with individual parameter $\delta^{(h)}$ to achieve hierarchy-wise semantic segmentation.

We further give Theorem 1 which indicates that our late-decoupled decoders can mitigate gradient conflicts and achieve per-hierarchy minima. This design makes the foundational knowledge in the shared encoder $\mathcal{E}_\theta$ transfer efficiently to all hierarchies, while ensuring the decoupled decoders $\{\mathcal{G}_{\delta^{(h)}}^{(h)}\}_{h=1}^H$ can mitigate the negative conflicts among multi-hierarchy semantic segmentation tasks.

**Theorem 1.** *Consider a multi-label learning problem with $H$ hierarchies of labels $\{\mathcal{L}^{(h)}\}_{h=1}^H$, where $\hat{\mathcal{L}}^{(h)}$ denotes the empirical risk at hierarchy $h$. Let $\delta$ denote shared decoder parameters across all hierarchies, and $\delta^{(h)}$ denote late-decoupled decoder parameters for each hierarchy. Define*

$$\delta^* = \arg\min_\delta \sum_{h=1}^H \hat{\mathcal{L}}^{(h)}(\theta, \delta), \ \delta^{*(h)} = \arg\min_{\delta^{(h)}} \hat{\mathcal{L}}^{(h)}(\theta, \delta^{(h)}).$$

*Then $\hat{\mathcal{L}}^{(h)}(\theta, \delta^{*(h)}) \leq \hat{\mathcal{L}}^{(h)}(\theta, \delta^*), \forall h$, with strict inequality for at least one hierarchy. Consequently, we have*

$$\sum_{h=1}^H \hat{\mathcal{L}}^{(h)}(\theta, \delta^{*(h)}) \leq \sum_{h=1}^H \hat{\mathcal{L}}^{(h)}(\theta, \delta^*),$$

*which is expected to outperform the shared decoders both per-hierarchy and in aggregate empirical risk.*

*Proof.* Proof is given in Appendix A. $\square$

**Coarse-to-fine hierarchical guidance.** To fully utilize the multi-hierarchy discriminability, we propose the coarse-to-fine hierarchical guidance which fuses the semantic prediction of one hierarchy with the features of the next hierarchy, enabling their information flow and mutual enhancement.

To be specific, we use $\mathbf{H}^{(h-1)}$ to represent the middle feature between $\mathbf{Y}^{(h-1)}$ and $\mathbf{Z}$. Then, the coarse-to-fine hierarchical guidance module can be formulated as follows:

$$\hat{\mathbf{H}}^{(h)} = \mathrm{MLP}\left(\left[\mathbf{H}^{(h)} \oplus \alpha \cdot \mathrm{MLP}(\mathbf{Y}^{(h-1)})\right]\right), \quad (3)$$

where MLP denotes the multi-layer perceptron, $\oplus$ denotes the concatenate operation, and $\alpha$ is a trade-off parameter to balance the influence between two hierarchies. Then, the semantic prediction for the $h$-th hierarchy is obtained by

$$\mathbf{Y}^{(h)} = \mathrm{Classifier}(\hat{\mathbf{H}}^{(h)}). \quad (4)$$

The classifier module projects the fused feature $\hat{\mathbf{H}}^{(h)}$ to the probabilistic prediction $\mathbf{Y}^{(h)}$. In this way, the late-decoupled 3DHS branch is trained by cross-entropy loss:

$$\mathcal{L}_{\mathrm{ces}} = \sum_{h=1}^H \mathcal{L}^{(h)} = -\sum_{h=1}^H \frac{1}{N} \sum_{i=1}^N \sum_{j=1}^{K^{(h)}} \hat{y}_{i,j}^{(h)} \log(y_{i,j}^{(h)}). \quad (5)$$

$\hat{y}_{i,j}^{(h)} \in \hat{\mathbf{Y}}^{(h)}$ is the one-hot ground-truth label (1 if point $i$ belongs to class $j$, 0 otherwise), and $y_{i,j}^{(h)} \in \mathbf{Y}^{(h)}$ is the model's output probability for point $i$ on class $j$.

**Cross-hierarchy consistency.** To take advantage of the parent-child structures (Athanasopoulos et al., 2024; Li et al., 2025) within hierarchical labels in Eq. (5), we further establish the cross-hierarchy consistency loss for optimizing the late-decoupled 3DHS network. Concretely, we define a mapping matrix $\mathbf{A}^{(h,h-1)} \in \{0,1\}^{K^{(h)} \times K^{(h-1)}}$ which indicates the parent-child relationship between the $(h-1)$-th and $h$-th hierarchies. The mapping matrix can convert coarse-grained labels into fine-grained labels, and its formal definition can be written as

$$\min_{\mathbf{A}^{(h,h-1)}} \|\hat{\mathbf{Y}}^{(h)} - \hat{\mathbf{Y}}^{(h-1)}(\mathbf{A}^{(h,h-1)})^T\|_F^2, \quad (6)$$

where $(\cdot)^T$ is the matrix transposition operation, $\hat{\mathbf{Y}}^{(h)} \in \{0,1\}^{N \times K^{(h)}}$ and $\hat{\mathbf{Y}}^{(h-1)} \in \{0,1\}^{N \times K^{(h-1)}}$ denote the one-hot label matrices. Then, the following loss is employed to achieve our cross-hierarchy consistency:

$$\mathcal{L}_{\mathrm{chc}} = \frac{1}{N} \sum_{i=1}^N \sum_{h=2}^H \left\|\mathbf{y}_i^{(h)} - \mathbf{A}^{(h,h-1)} \mathbf{y}_i^{(h-1)}\right\|_2^2, \quad (7)$$

where $\mathbf{y}_i^{(h)}$ is the prediction vector for the point $i$ at the hierarchy level $h$ with a shape of $K^{(h)} \times 1$. Finally, the objective for training our late-decoupled 3DHS branch combines the cross-entropy and the cross-hierarchy consistency losses as follows:

$$\mathcal{L}_{\mathrm{Ld\text{-}3DHS}} = \mathcal{L}_{\mathrm{ces}} + \mathcal{L}_{\mathrm{chc}}. \quad (8)$$

## 3.3. 3DHS-oriented contrastive cluster self-supervision learning

In this paper, we further introduce the contrastive cluster self-supervision learning to mitigate the class-imbalance issue in 3DHS as shown in Figure 1(b). Specifically, in the training process, we divide the complete scene point cloud $\mathbf{X}$ into multiple cluster-wise point clouds $\{\mathbf{X}_i^{(h)}\}_{i=1}^{K^{(h)}}$, where $\mathbf{X}_i^{(h)}$ denotes the $i$-th class data in the $h$-th hierarchy. Then, we employ contrastive learning to learn cluster-wise discriminative features in the *contrastive discrimination branch* and further construct semantic cluster prototypes, to provide helpful self-supervision signals for improving the *late-decoupled multi-label learning 3DHS branch*.

**Contrastive discrimination.** Given $\{\mathbf{X}^{(h,c)}\}_{c=1}^{K^{(h)}}$ ($c$ denotes the $c$-th class), we first utilize the point cloud encoder to extract class-wise features and stack a MLP projection head to obtain contrastive features $\{\mathbf{F}^{(h,c)}\}_{c=1}^{K^{(h)}}$. Then, we minimize the supervised contrastive loss (Khosla et al., 2020) to achieve the feature discrimination objective:

$$\mathcal{L}_{\mathrm{con}}^{(h)} = -\mathbb{E}_{s^+ \in \mathcal{P}^{(h)}} \left[ s^+ - \log \sum\nolimits_{s^- \in \mathcal{N}^{(h)}} e^{s^-} \right], \quad (9)$$

where $\mathcal{P}^{(h)}$ and $\mathcal{N}^{(h)}$ denote the sets of positive and negative pairs, respectively. If the features $\mathbf{f}_i^{(h,c_1)} \in \mathbf{F}^{(h,c_1)}$ and $\mathbf{f}_j^{(h,c_2)} \in \mathbf{F}^{(h,c_2)}$ come from the same class (i.e., $c_1 = c_2$) in the $h$-th hierarchy, their feature pair belongs to $\mathcal{P}^{(h)}$, otherwise the pair belongs to $\mathcal{N}^{(h)}$. $s^+$ and $s^-$ measure the feature distance computed by cosine similarity.

This discrimination objective is expected to pull the intra-class features closer together while pushing inter-class features further apart, thereby learning discriminative point cloud features for individual classes.

**Semantic clusters for bi-branch self-supervision.** Given the class-wise discriminative features $\{\mathbf{F}^{(h,c)}\}_{c=1}^{K^{(h)}}$ in the contrastive discrimination branch, we use the same manner to divide the middle features $\mathbf{H}^{(h)}$ into class-wise middle features $\{\mathbf{H}^{(h,c)}\}_{c=1}^{K^{(h)}}$ in the late-decoupled 3DHS branch. Then, we propose the semantic-cluster-based bi-branch self-supervision mechanism to promote class-imbalance 3DHS tasks. Formally, for the $h$-th hierarchy, we compute semantic cluster prototypes on both branches using the feature expectation of each class:

$$\mathbf{p}_{\mathrm{3D}}^{(h,c)} = \mathbb{E}_i[\mathbf{h}_i^{(h,c)}], \ \mathbf{p}_{\mathrm{aux}}^{(h,c)} = \mathbb{E}_i[\mathbf{f}_i^{(h,c)}], \quad (10)$$

where $\mathbf{h}_i^{(h,c)} \in \mathbf{H}^{(h,c)}, \mathbf{f}_i^{(h,c)} \in \mathbf{F}^{(h,c)}$ denote the $i$-th sample features in the $c$-th class. $\mathbf{p}_{\mathrm{3D}}^{(h,c)}$ and $\mathbf{p}_{\mathrm{aux}}^{(h,c)}$ are defined as the semantic cluster prototypes that extract the general class information from the main 3DHS branch and the auxiliary contrastive discrimination branch. Furthermore, we leverage the two semantic cluster prototypes to mutually

supervise two branches by the following loss:

$$\mathcal{L}_{\mathrm{bis}}^{(h)} = \sum_c \frac{1}{N^{(h,c)}} \sum_i \left[ \mathrm{Smooth}_{L1} \left( \mathbf{p}_{\mathrm{3D}}^{(h,c)} - \mathbf{f}_i^{(h,c)} \right) \right.$$
$$\left. + \mathrm{Smooth}_{L1} \left( \mathbf{p}_{\mathrm{aux}}^{(h,c)} - \mathbf{h}_i^{(h,c)} \right) \right], \quad (11)$$

where $N^{(h,c)}$ is the feature number of the $c$-th class in the $h$-th hierarchy. $\mathrm{Smooth}_{L1}$ denotes the smooth L1 loss which is commonly employed to increase the robustness to outliers (Girshick, 2015), satisfying that $\mathrm{Smooth}_{L1}(x) = 0.5x^2$, if $|x| < 1$, otherwise $\mathrm{Smooth}_{L1}(x) = |x| - 0.5$.

In our auxiliary contrastive discrimination branch, the contrastive loss $\mathcal{L}_{\mathrm{con}}^{(h)}$ and the bi-branch self-supervised loss $\mathcal{L}_{\mathrm{bis}}^{(h)}$ are combined for joint optimization:

$$\mathcal{L}_{\mathrm{aux}}^{(h)} = \mathcal{L}_{\mathrm{con}}^{(h)} + \mathcal{L}_{\mathrm{bis}}^{(h)}. \quad (12)$$

Overall, the entire loss in our method is formulated as:

$$\mathcal{L}_{\mathrm{total}} = \mathcal{L}_{\mathrm{Ld\text{-}3DHS}} + \lambda \sum_{h=1}^{H} \mathcal{L}_{\mathrm{aux}}^{(h)}, \quad (13)$$

where $\lambda$ is a hyper-parameter to balance the two branches' training. In this way, our model is expected to leverage the discriminative class-wise information in the auxiliary branch to promote the hierarchical semantic segmentation tasks for rare classes in the late-decoupled 3DHS branch. To ensure the balance between the effectiveness and efficiency, in experiments, we establish a Gini coefficient (Dorfman, 1979) based activation strategy for the auxiliary branch. The details are shown in Appendix B.4.

## 4. Experiments

### 4.1. Experimental setup

**Datasets.** We conduct experiments on three datasets featuring hierarchical annotations. Their details are as follows:

**1) Campus3D:** The Campus3D (Li et al., 2020) dataset natively provides 5 semantic hierarchies. In our experiments, we select the representative 1-st, 3-rd, and 5-th hierarchies to form a 3-hierarchy evaluation benchmark. Following (Li et al., 2025), to handle extreme sparsity at the finest granularity, the three least common classes are merged into a single 'miscellaneous' category. We follow the standard evaluation process, i.e., training on areas FASS, YIH, RA, UCC, validating on PGP, and testing on FOE.

**2) S3DIS-H:** The Stanford 3D Indoor Spaces (S3DIS (Armeni et al., 2016)) dataset consists of 6 large indoor areas with 13 semantic categories. As this dataset lacks native hierarchical labels, we constructed a 2-hierarchy by following the automated pipeline (Li et al., 2025), which we term S3DIS-H and adhere to the standard evaluation protocol of training on Areas 1-4, 6 and testing on Area 5.

*Table 1.* **3D hierarchical semantic segmentation on three datasets.** Results of mIoU (%) of each hierarchy (L0∼L2) and their average results (Avg.mIoU) are reported. The Avg.mIoU improvement achieved by our ML3DHS over the best prior method is shown in red.

| Backbone | Method | Campus3D | | | | S3DIS-H | | | SensatUrban-H | | |
|---|---|---|---|---|---|---|---|---|---|---|---|
| | | Avg. mIoU | L0 | L1 | L2 | Avg. mIoU | L0 | L1 | Avg. mIoU | L0 | L1 |
| PointNet++ | MTHS (Li et al., 2020) | 58.80 | 92.90 | 42.50 | 41.00 | 62.74 | 66.95 | 58.52 | 47.20 | 51.89 | 42.51 |
| | DHL (Li et al., 2025) | 62.56 | 91.68 | **54.89** | 41.12 | 63.05 | 67.45 | 58.65 | 48.20 | **57.20** | 39.20 |
| | **ML3DHS (ours)** | **63.28** (+0.72) | **93.02** | 52.22 | **44.61** | **66.43** (+3.38) | **70.06** | **62.80** | **49.73** (+1.53) | 54.79 | **44.67** |
| Point TF v2 | MTHS (Li et al., 2020) | 63.85 | 92.40 | 56.14 | 43.01 | 74.90 | 78.56 | 71.23 | 49.27 | 54.22 | 44.32 |
| | DHL (Li et al., 2025) | 65.80 | 92.72 | 60.36 | 44.31 | 75.76 | 79.66 | 71.86 | 50.62 | 55.62 | 45.62 |
| | **ML3DHS (ours)** | **66.87** (+1.07) | **93.05** | **61.85** | **45.71** | **76.71** (+0.95) | **80.30** | **73.12** | **53.56** (+2.94) | **58.98** | **48.14** |
| Point TF v3 | MTHS (Li et al., 2020) | 64.96 | 92.90 | 57.83 | 44.15 | 71.54 | 75.59 | 67.48 | 39.07 | 42.90 | 35.24 |
| | DHL (Li et al., 2025) | 62.90 | 91.20 | 55.50 | 42.00 | 72.83 | 76.32 | 69.34 | 39.41 | 43.29 | 35.53 |
| | **ML3DHS (ours)** | **66.40** (+1.44) | **93.40** | **60.50** | **45.30** | **74.50** (+1.67) | **78.30** | **70.70** | **40.43** (+1.02) | **44.42** | **36.43** |

*Table 2.* Performance gain (Avg. mIoU) before and after applying our ML3DHS to MTHS and DHL using PointNet++.

| Method | Dataset | | |
|---|---|---|---|
| | Campus3D | S3DIS-H | SensatUrban-H |
| MTHS (baseline) | 58.80 | 62.74 | 47.20 |
| **MTHS + ML3DHS** | **59.96** (+1.16) | **63.93** (+1.19) | **49.05** (+1.85) |
| DHL (baseline) | 62.56 | 63.05 | 48.20 |
| **DHL + ML3DHS** | **64.29** (+1.73) | **63.59** (+0.54) | **49.12** (+0.92) |

*Table 3.* Performance gain (Avg. mIoU) before and after applying our ML3DHS to MTHS and DHL using Point Transformer v2.

| Method | Dataset | | |
|---|---|---|---|
| | Campus3D | S3DIS-H | SensatUrban-H |
| MTHS (baseline) | 63.85 | 74.90 | 49.27 |
| **MTHS + ML3DHS** | **66.38** (+2.53) | **76.40** (+1.50) | **51.07** (+1.80) |
| DHL (baseline) | 65.80 | 75.76 | 50.62 |
| **DHL + ML3DHS** | **66.94** (+1.14) | **76.76** (+1.00) | **53.61** (+2.99) |

*Table 4.* Performance gain (Avg. mIoU) before and after applying our ML3DHS to MTHS and DHL using Point Transformer v3.

| Method | Dataset | | |
|---|---|---|---|
| | Campus3D | S3DIS-H | SensatUrban-H |
| MTHS (baseline) | 64.96 | 71.54 | 39.07 |
| **MTHS + ML3DHS** | **66.12** (+1.16) | **74.20** (+2.66) | **40.12** (+1.05) |
| DHL (baseline) | 62.90 | 72.83 | 39.41 |
| **DHL + ML3DHS** | **64.67** (+1.77) | **75.34** (+2.51) | **40.44** (+1.03) |

**3) SensatUrban-H:** The SensatUrban-H dataset is a 2-hierarchy large-scale urban benchmark. This version was constructed by (Li et al., 2025) from the original SensatUrban (Hu et al., 2022) dataset, and we directly utilize their publicly released version, which partitions the dataset's blocks from the two cities, i.e., Birmingham and Cambridge, into distinct training, validation, and test sets.

**Comparison baselines.** We compare our method against the classical and the state-of-the-art 3DHS methods: MTHS (Li et al., 2020) and DHL (Li et al., 2025). To ensure a fair comparison that isolates the contribution of our methodology from backbone differences, our evaluation is on multiple backbones including PointNet++ (Qi et al., 2017b), Point Transformer v2 (Wu et al., 2022) (Point TF v2), and v3 (Wu et al., 2024a) (Point TF v3). This allows a comprehensive evaluation of method effectiveness.

**Evaluation metrics.** The primary evaluation metric for all experiments is the mean Intersection over Union (mIoU). Crucially, to provide a comprehensive assessment of 3DHS methods, we report the mIoU score for each hierarchy separately and also report the average mIoU (Avg.mIOU) across all hierarchies for an overall performance summary.

**Implementation details.** More experimental details are given in Appendix B. The implementation code is available at https://github.com/MideasternM/ML3DHS.

**4.2. Quantitative comparison.**

**Comparison experiments.** We comprehensively compare our method ML3DHS with the representative 3DHS methods MTHS and DHL across three datasets based on three backbones. The results are listed in Table 1, from which we can observe that: 1) Our ML3DHS consistently achieves superior 3DHS performance across all tested datasets. For instance, with the same backbone PointNet++, the Avg.mIoU of ML3DHS surpasses that of DHL by **3.38%** on indoor benchmark S3DIS-H and **1.53%** on outdoor benchmark SensatUrban-H. 2) Our ML3DHS is compatible with different backbones and has achieved new SOTA results. For example, on Campus3D, our ML3DHS outperforms the highly competitive DHL by **0.72%**, **1.07%**, and **1.44%** Avg.mIoU using PointNet++, Point TF v2, and Point TF v3, respectively. These improvements indicate that our ML3DHS is able to achieve better segmentation performance at multiple hierarchies.

**Add-on experiments.** In Tables 2, 3, and 4, we conduct add-on experiments by applying our ML3DHS to previous methods MTHS and DHL. Specifically, we employ the model structure and loss function of our ML3DHS to the methods MTHS and DHL to reconstruct their variants (i.e., MTHS+ML3DHS, DHL+ML3DHS), and then re-run their

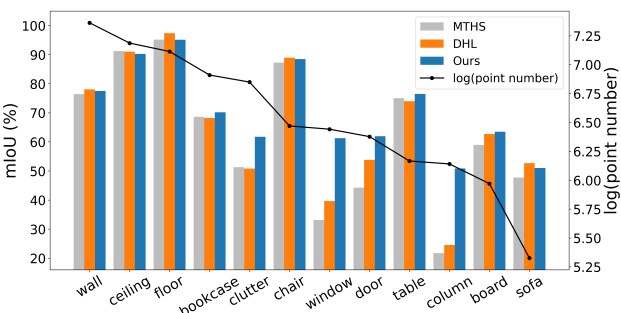

*Figure 2.* Per-class segmentation performance comparison of three methods on S3DIS-H dataset. The mIoU values of the L1 hierarchy on S3DIS-H dataset are reported, and the black line records the log-scale point number for individual classes.

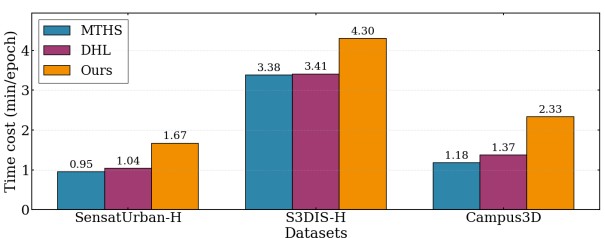

*Figure 3.* Time cost comparison of three 3DHS methods.

algorithms. The results indicate that ML3DHS can excel as an auxiliary module to boost the performance of previous methods. For example, on SensatUrban-H dataset with PointNet++ backbone, our ML3DHS yields **+1.85%** improvement of Avg.mIoU when it is integrated with the MTHS baseline. On Campus3D dataset with Point TF v3 backbone, ML3DHS contributes to an Avg.mIoU gain of **+1.77%**. As a consequence, applying our ML3DHS consistently improves the hierarchical segmentation results across different datasets, backbones, and baseline methods, thus further validating our method's effectiveness.

### 4.3. Qualitative results

**Segmentation performance** *vs.* **point number.** In Figure 2, we visualize the segmentation performance on each class and the corresponding point number on the finest hierarchy of S3DIS-H. Compared with MTHS and DHL, our method achieves comparable performance on the majority classes, such as *wall*, *floor*, *ceiling*, *table*, and *chair*. Notably, our method achieves significant performance improvement on the minority classes, such as *clutter*, *window*, *door*, and *column*. These trends in Figure 2 are obvious and they demonstrate the effectiveness of our proposed method in addressing the class imbalance issue in 3DHS tasks.

**Time cost comparison for 3DHS.** To evaluate the time cost of MTHS, DHL, and our ML3DHS, we record their average time per epoch in training stages on the same de-

*Table 5.* **Ablation study on model structure.** Results of Avg. mIoU (%) are reported. LDF: late-decoupled framework, CFG: coarse-to-fine guidance, CDB: contrastive discrimination branch.

| Model variant | | | Dataset | | |
| --- | --- | --- | --- | --- | --- |
| LDF | CFG | CDB | Campus3D | S3DIS-H | SensatUrban-H |
| ✓ | ✓ | ✓ | 63.28 | 66.43 | 49.73 |
| ✗ | ✓ | ✓ | 59.22 (-4.06) | 62.17 (-4.26) | 47.74 (-1.99) |
| ✓ | ✗ | ✓ | 59.96 (-3.32) | 63.93 (-2.50) | 49.05 (-0.68) |
| ✓ | ✓ | ✗ | 61.95 (-1.33) | 62.98 (-3.45) | 48.15 (-1.58) |

*Table 6.* **Ablation study on loss components.** Results of Avg. mIoU (%) are reported. $\mathcal{L}_{con}$: contrastive loss, $\mathcal{L}_{chc}$: cross-hierarchy consistency loss, $\mathcal{L}_{bis}$: bi-branch self-supervised loss.

| Loss component | | | Dataset | | |
| --- | --- | --- | --- | --- | --- |
| $\mathcal{L}_{con}$ | $\mathcal{L}_{chc}$ | $\mathcal{L}_{bis}$ | Campus3D | S3DIS-H | SensatUrban-H |
| ✓ | ✓ | ✓ | 63.28 | 66.43 | 49.73 |
| ✗ | ✓ | ✓ | 59.15 (-4.13) | 61.49 (-4.94) | 48.75 (-0.98) |
| ✓ | ✗ | ✓ | 61.08 (-2.20) | 64.09 (-2.34) | 47.50 (-2.23) |
| ✓ | ✓ | ✗ | 61.95 (-1.33) | 62.98 (-3.45) | 48.15 (-1.58) |

vice. The results on three datasets using PointNet++ backbone are shown in Figure 3 which indicates the time consumption of our method is slightly higher than that of the comparison ones. This is mainly because our method needs to train the auxiliary contrastive discrimination branch to improve segmentation performance. Overall, our method achieves a more robust 3DHS at an affordable time cost.

**Case comparison for 3DHS.** Figure 4 provides a qualitative comparison of our method against MTHS and DHL on the challenging indoor benchmark S3DIS-H. It shows that all three methods have satisfactory performance at the coarse-grained point cloud segmentation, e.g., at the L0 hierarchy as shown in the visualization cases. However, we can observe the failure cases at the fine-grained point cloud segmentation (L1 hierarchy) of previous 3DHS methods. For example, as indicated by the red circles in MTHS column, MTHS mis-segments the points of wall and window compared to the ground truth. As indicated by the blue circles in DHL column, DHL mis-segments the points at the corner and boundary compared to the ground truth. In contrast, our method successfully overcomes the challenging segmentation tasks for those point clouds. As highlighted in the red and blue circles in the last column, our method correctly segments the points (which are wrongly segmented by MTHS and DHL) compared with the ground truth, indicating its ability to perform fine-grained point cloud segmentation. These visualization results verified the effectiveness of our method again.

### 4.4. Ablation studies

In this part, we conduct ablation studies to validate the effectiveness of key components in our framework. Tables 5

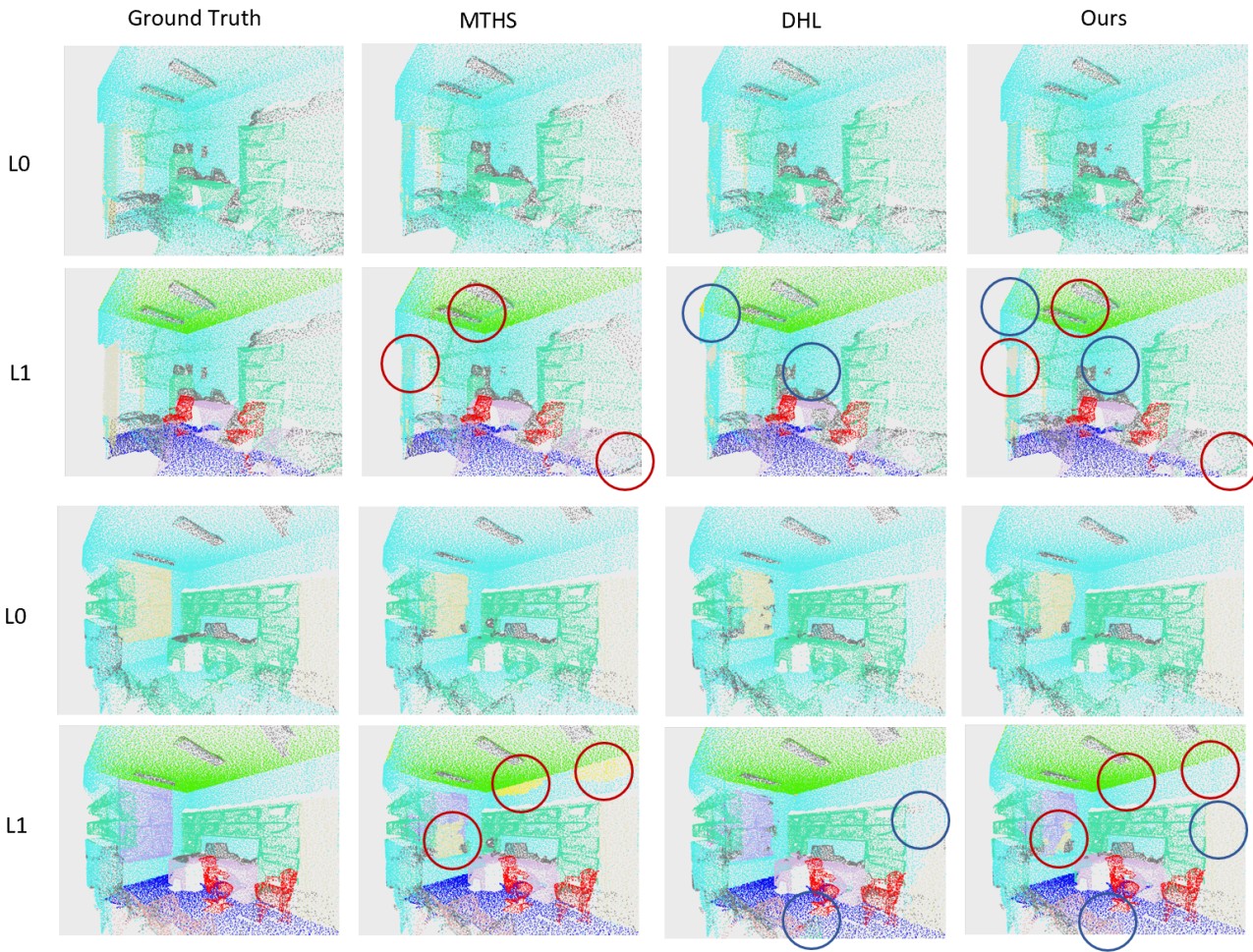

*Figure 4.* Case comparison of three 3D hierarchical semantic segmentation methods (MTHS (Li et al., 2020), DHL (Li et al., 2025), and our ML3DHS) on testing samples of S3DIS-H. L0 and L1 represent the two semantic hierarchies in the indoor point cloud scenes.

and 6 report the results on three datasets using PointNet++.

**Ablation study on model structure.** From the perspective of model structure, the core designs in our framework include the late-decoupled framework (LDF), coarse-to-fine guidance (CFG), and contrastive discrimination branch (CDB). Table 5 shows their ablation experiments. First, removing LDF results in the most severe performance drop, e.g., the Avg.mIoU significantly decreases by $4.06\%$ on Campus3D and $4.26\%$ on S3DIS-H, respectively. This demonstrates the crucial effect of our proposed late-decoupled framework in mitigating multi-hierarchy conflicts. Second, disabling CFG also leads to a noticeable $3.32\%$ performance drop on Campus3D, validating the efficacy of using coarser hierarchies to guide the fine hierarchies. Third, the removal of CDB also causes performance degradation on all datasets (e.g., $3.45\%$ drop on S3DIS-H), indicating our contrastive discrimination branch is helpful for the fine-grained 3DHS tasks with class imbalance.

**Ablation study on loss components.** From the perspective of optimization loss, the key components in our framework include the contrastive loss ($\mathcal{L}_{con}$), cross-hierarchy consistency loss ($\mathcal{L}_{chc}$), and bi-branch self-supervised loss ($\mathcal{L}_{bis}$). Table 6 shows their ablation experiments. Specifically, the contribution of $\mathcal{L}_{con}$ was particularly significant, e.g., its removal incurs a substantial Avg.mIoU drop of $4.94\%$ on the S3DIS-H dataset. Similarly, ablating $\mathcal{L}_{chc}$ also results in performance degeneration, e.g., $2.34\%$ Avg.mIoU drop on S3DIS-H. The removal of $\mathcal{L}_{bis}$ also causes a clear performance drop across all three datasets, i.e., a $1.33\%$ decrease on Campus3D, $3.45\%$ on S3DIS-H, and $1.58\%$ on SensatUrban-H. These results demonstrate the necessary role of each component in our ML3DHS framework.

**Hyper-parameter analysis.** Our method includes two hyper-parameters: the weight $\alpha$ in coarse-to-fine guidance in Eq. (3) and the weight $\lambda$ of auxiliary branch loss in Eq. (13). The parameter analysis is shown in Figure 5. For $\alpha$, the performance peaks in a moderate range, i.e., $0.1 < \alpha < 10.0$. For $\lambda$, we find it is insensitive at the range

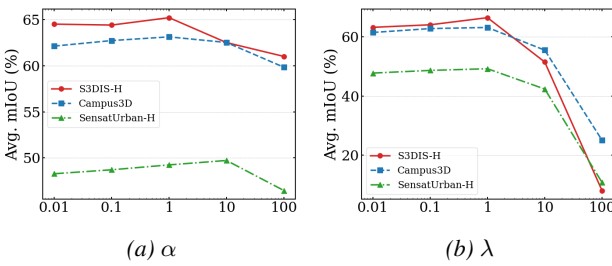

*Figure 5.* Hyper-parameter analysis: (a) coarse-to-fine guidance weight $\alpha$, (b) contrastive discrimination branch loss weight $\lambda$.

of $[0.1, 1.0]$. For generality, we set $\alpha = 1.0$ and $\lambda = 1.0$ for comparison experiments on all datasets and backbones.

## 5. Conclusion

In this paper, we focus on addressing two persistent but overlooked challenges for 3D hierarchical semantic segmentation: the multi-hierarchy conflict between learning multiple labels and a parameter-sharing model, and the class imbalance issue hindering performance on minority classes. To this end, we propose ML3DHS, a novel late-decoupled multi-label learning framework with the contrastive cluster self-supervision method. The late-decoupled 3DHS architecture can mitigate the multi-hierarchy conflicts and the auxiliary contrastive discrimination branch can improve the segmentation ability for point clouds with class imbalance. Extensive experiments on multiple datasets and backbones demonstrate that our approach achieves state-of-the-art 3DHS performance, whose core components can also be used as a plug-and-play enhancement to improve previous 3DHS methods.

**Limitation and future work.** Although our ML3DHS is proven to be effective, the non-shared decoders and the extra auxiliary branch will increase the model size for training. Therefore, future work can explore lightweight models and open-vocabulary strategies to further improve the efficiency and performance on extremely rare classes.

## Impact Statement

This paper presents work whose goal is to advance the field of machine learning and 3D computer vision. There are many potential societal consequences of our work, none of which we feel must be specifically highlighted here.

## Acknowledgments

This research was supported by the Ministry of Education, Singapore, under its MOE Academic Research Fund Tier 2 (MOE-T2EP20124-0013), and Sichuan Science and Technology Program (Grant No: 2026YFHZ0077).

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

# Appendix

## A. Theoretical analysis

*Proof.* We now provide a step-by-step justification for our Theorem 1 by using convex analysis, gradient conflict arguments, and Pareto optimality.

**Step 1: Independent minimizers.** By definition,

$$\delta^{*(h)} = \arg\min_{\delta^{(h)}} \hat{\mathcal{L}}^{(h)}(\theta, \delta^{(h)}).$$

Hence,

$$\hat{\mathcal{L}}^{(h)}(\theta, \delta^{*(h)}) \leq \hat{\mathcal{L}}^{(h)}(\theta, \delta), \quad \forall \delta.$$

In particular, substituting $\delta = \delta^*$ yields

$$\hat{\mathcal{L}}^{(h)}(\theta, \delta^{*(h)}) \leq \hat{\mathcal{L}}^{(h)}(\theta, \delta^*), \quad \forall h.$$

**Step 2: Shared decoder optimality condition.** The shared solution $\delta^*$ minimizes the scalarized objective

$$J(\delta) = \sum_{h=1}^{H} \hat{\mathcal{L}}^{(h)}(\theta, \delta).$$

Thus the first-order condition is

$$\sum_{h=1}^{H} \nabla_\delta \hat{\mathcal{L}}^{(h)}(\theta, \delta^*) = 0.$$

This balances gradients across hierarchies but does not imply

$$\nabla_\delta \hat{\mathcal{L}}^{(h)}(\theta, \delta^*) = 0, \quad \forall h.$$

Therefore, $\delta^*$ is not the minimizer of each hierarchy's risk.

**Step 3: Gradient conflict analysis.** Define

$$g^{(h)}(\delta) := \nabla_\delta \hat{\mathcal{L}}^{(h)}(\theta, \delta).$$

At $\delta^*$ we have $\sum_h \lambda^{(h)} g^{(h)}(\delta^*) = 0$. Suppose there exist $h_1, h_2$ such that

$$\langle g^{(h_1)}(\delta^*), g^{(h_2)}(\delta^*) \rangle < 0.$$

Then moving in the direction $-g^{(h_1)}(\delta^*)$ yields

$$\frac{d}{d\epsilon} \hat{\mathcal{L}}^{(h_1)}(\theta, \delta^* - \epsilon g^{(h_1)}(\delta^*))\Big|_{\epsilon=0} = -\|g^{(h_1)}(\delta^*)\|^2 < 0,$$

but simultaneously

$$\frac{d}{d\epsilon} \hat{\mathcal{L}}^{(h_2)}(\theta, \delta^* - \epsilon g^{(h_1)}(\delta^*))\Big|_{\epsilon=0} = -\langle g^{(h_2)}(\delta^*), g^{(h_1)}(\delta^*) \rangle > 0.$$

Thus improving hierarchy $h_1$ necessarily worsens hierarchy $h_2$. This shows $\delta^*$ is a compromise solution, not a per-hierarchy minimizer.

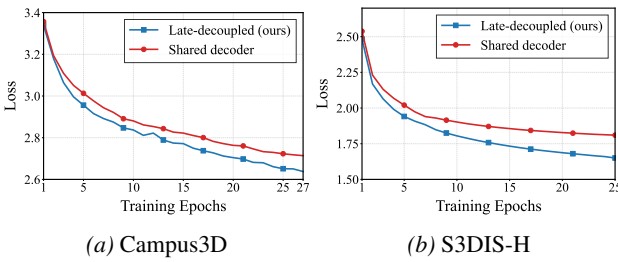

*(a)* Campus3D      *(b)* S3DIS-H

*Figure 6.* Loss comparison on (a) Campus3D and (b) S3DIS-H. Our late-decoupled multi-label learning model achieves lower cross-entropy loss of all hierarchies and faster convergence than the shared decoder model, which is aligned with our Theorem 1.

**Step 4: Pareto frontier characterization.** Define the multi-objective vector

$$F(\delta) = \big(\hat{\mathcal{L}}^{(1)}(\theta, \delta), \dots, \hat{\mathcal{L}}^{(H)}(\theta, \delta)\big).$$

A point $\delta$ is Pareto optimal if no $\delta'$ exists such that

$$\hat{\mathcal{L}}^{(h)}(\theta, \delta') \leq \hat{\mathcal{L}}^{(h)}(\theta, \delta), \quad \forall h,$$

with strict inequality for some $h$. Each $\delta^{*(h)}$ lies on the Pareto frontier, since it minimizes one objective exactly. By contrast, $\delta^*$ minimizes only the scalarization $J(\delta)$, which may lie strictly inside the feasible region. If $\delta^*$ were Pareto optimal for all hierarchies, then

$$g^{(h)}(\delta^*) = 0, \quad \forall h,$$

contradicting Step 2 unless all risks share identical minimizers. Hence $\delta^*$ is generally Pareto suboptimal.

**Step 5: Second-order stability.** At $\delta^*$, the Hessian of $J$ is

$$\nabla_\delta^2 J(\delta^*) = \sum_{h=1}^{H} \nabla_\delta^2 \hat{\mathcal{L}}^{(h)}(\theta, \delta^*).$$

Even if this aggregate Hessian is positive semidefinite, individual Hessians may not be minimized at $\delta^*$. Thus $\delta^*$ can be a saddle point for some hierarchy $h$, while $\delta^{*(h)}$ is a strict minimizer.

**Step 6: Strict inequality and summation.** If $\delta^*$ were optimal for all hierarchies, then $g^{(h)}(\delta^*) = 0$ for all $h$, contradicting Step 2 unless all objectives share the same minimizer. Therefore, there exists $h_0$ such that

$$\hat{\mathcal{L}}^{(h_0)}(\theta, \delta^{*(h_0)}) < \hat{\mathcal{L}}^{(h_0)}(\theta, \delta^*).$$

Summing across all hierarchies yields

$$\sum_{h=1}^{H} \hat{\mathcal{L}}^{(h)}(\theta, \delta^{*(h)}) \leq \sum_{h=1}^{H} \hat{\mathcal{L}}^{(h)}(\theta, \delta^*),$$

with strict inequality by existence of $h_0$. □

## B. Implementation details

**Summary:** We employ an adaptive decoder strategy to balance performance and efficiency. To be specific, for the lightweight backbone PointNet++, we use multiple copies of the original decoder to implement the late-decoupled decoders. For Point Transformer v2 and v3, multiple lightweight MLPs are used as late-decoupled decoders to achieve efficient 3DHS. All experiments are conducted on a single NVIDIA RTX 4090 GPU (24GB) using PyTorch v2.0.0 with CUDA 11.8. Models were trained for 100 epochs using the AdamW optimizer with betas of (0.9, 0.999) and a weight decay of 1e-4. The learning rate is managed by a cosine annealing scheduler, starting at 0.01 and decaying to 1.0e-5. The batch size is set to 12 for S3DIS-H and 64 for Campus3D and SensatUrban-H. Following common practices in 3D segmentation, we leverage a series of online data augmentations to train the model, including random scaling within the $[0.9, 1.1]$ range, random point jittering with a Gaussian noise of $\sigma = 0.005$, and a 20% probability of color dropping. In all our comparison experiments, all hyper-parameters for weighting the different loss components are set to $1.0$.

### B.1. KEY HYPER-PARAMETERS

Table 7 summarizes the key hyper-parameters used in our ML3DHS framework.

*Table 7.* Key hyper-parameters in ML3DHS.

| Hyper-parameter | Symbol | Value |
|---|---|---|
| EMA momentum | $\beta$ | 0.999 |
| Auxiliary loss weight | $\lambda$ | 1.0 |
| Guidance weight | $\alpha$ | 1.0 |
| Contrastive temperature | $\tau$ | 0.07 |
| Learning rate (initial) | $\eta$ | 0.01 |
| Learning rate (final) | - | 1.0e-5 |
| Weight decay | - | 1.0e-4 |
| Batch size (S3DIS) | $|B|$ | 12 |
| Batch size (Campus, Urban) | $|B|$ | 64 |
| Label smoothing factor | - | 0.2 |
| Training epochs | - | 100 |

### B.2. NETWORK ARCHITECTURE

This section details the network architecture of our ML3DHS model. The structural specifications for the main and auxiliary branches are presented in Table 8.

### B.3. CONTRASTIVE LEARNING AND CLUSTER PROTOTYPE UPDATE

**Contrastive loss:** Temperature $\tau = 0.07$. Features are L2-normalized. Negative samples: all features from other classes in the same batch.

**Cluster update:** The cluster prototypes for each class are

*Table 8.* Network architecture of ML3DHS. $K^{(h)}$ denotes the number of classes in hierarchy $h$.

| Component | Specification |
|---|---|
| **Main Late-Decoupled 3DHS Branch** | |
| Encoder | **PointNet++** (4 Set Abstraction layers) 
 *MLP Dims:* $[32,32,64] \rightarrow [64,64,128] \rightarrow [128,128,256] \rightarrow [256,256,512]$ 
 *Local Grouping:* Radius $r \in \{0.1, 0.2, 0.4, 0.8\}$, max 32 points per region 
 *Downsampling:* Farthest Point Sampling (FPS) |
| Decoders ($\times H$) | **Feature Propagation** (4 layers) 
 *MLP Dims:* $[256,256] \rightarrow [256,256] \rightarrow [256,128] \rightarrow [128,128,128]$ 
 *Output:* 128-dim feature vector |
| Classifiers ($\times H$) | **MLP** with Dropout ($p = 0.5$) 
 *Dims:* $[128, 128, K^{(h)}]$ |
| **Auxiliary Contrastive Discriminative Branch** | |
| Encoder | **PointNet++** instance 
 *MLP Dims:* Same as the main branch encoder (with an additional stem convolution) 
 *Local Grouping:* Max 16 points per region (reduced for efficiency) |
| Projection Head | **MLP** 
 *Dims:* $[512, 256, 128]$ 
 *Output:* 128-dim feature vector for contrastive learning |

updated using Exponential Moving Average (EMA) with a high momentum coefficient $\beta = 0.999$, starting from the first epoch (no warm-up). This ensures a stable and smooth evolution of the class representations over time. The update rule for the semantic cluster $\mathbf{p}_{3D}^{(h,c)}$ (representing class $c$ in hierarchy $h$) is defined as:

$$\mathbf{p}_{3D}^{(h,c)} \leftarrow \beta \cdot \mathbf{p}_{3D}^{(h,c)} + (1 - \beta) \cdot \mathbf{p}_{new}^{(h,c)},$$

where $\mathbf{p}_{new}^{(h,c)}$ is the new cluster computed as the mean feature of all points belonging to class $c$ in the current mini-batch. These cluster prototypes are computed using ground truth labels and are L2-normalized after each update step.

### B.4. GINI COEFFICIENT BASED HIERARCHICAL IMBALANCE ACTIVATION

To adaptively manage class imbalance within each hierarchy, we employ a dynamic activation strategy based on the Gini coefficient. This strategy assesses if a hierarchy's class point distribution is severely imbalanced to determine whether to activate an imbalance handling mechanism (i.e., using the auxiliary contrastive discrimination branch with

loss $\mathcal{L}_{\text{aux}}$). Concretely, for hierarchy $h$, we quantify the imbalance by calculating the Gini coefficient $G_h$ from its class frequencies $f_{h,i}$:

$$G_h = \frac{\sum_{i=1}^{C_h} \sum_{j=1}^{C_h} |f_{h,i} - f_{h,j}|}{2 C_h \sum_{i=1}^{C_h} f_{h,i}}, \qquad (14)$$

where $C_h$ is the number of classes at hierarchy $h$, and $f_{h,i}$ is the point frequency of class $i$. The value of $G_h$ ranges from $0$ (perfect balance) to nearly $1$ (maximum imbalance).

We set an activation threshold $\gamma$ (e.g., 0.6). The imbalance handling module is activated only when $G_h \geq \gamma$, i.e., $\text{Activate}_{\text{Imbalance-h}} = (G_h \geq \gamma)$. This hierarchy-specific dynamic intervention allows the model to efficiently and selectively address severe class imbalance issues.

### B.5. DATASET HIERARCHY STRUCTURES

The hierarchical class structures, detailing the mapping from coarse to fine-grained categories for each dataset, are provided in Table 9. In our method, the mapping matrix $\mathbf{A}^{(h,h-1)}$, serving the cross-hierarchy consistency objective, is deterministic since class labels in each hierarchy are definite. Its closed-form solution can be expressed as:

$$\mathbf{A}^{(h,h-1)} = \left( \hat{\mathbf{Y}}^{(h-1)\dagger}, \hat{\mathbf{Y}}^{(h)} \right)^T,$$

where $\dagger$ denotes the pseudoinverse. We write the mapping matrix $\mathbf{A}^{(h,h-1)}$ in Eq. (7) for logical completeness. In practice, the label mapping is already pre-defined by the datasets as shown in Table 9. When class labels in the two hierarchies are definite, $\mathbf{A}^{(h,h-1)}$ is also determined. Hence, the matrix $\mathbf{A}^{(h,h-1)}$ is fixed during model training and it achieves the label mapping from the hierarchy $h$ to the hierarchy $(h-1)$.

### B.6. DETAILED ALGORITHMS

The Algorithm 1 outlines the overall training framework of our proposed ML3DHS. It demonstrates how the main branch (Late-decoupled 3DHS Branch) and the auxiliary branch (Contrastive Discrimination Branch) are jointly optimized in each training iteration.

## C. Additional analysis and experiments

We appreciate all the anonymous reviewers for their helpful comments during the rebuttal process of this work. In light of the comments, this section provides more experiments and analysis to support our method.

### C.1. COMPUTATIONAL EFFICIENCY ANALYSIS

We take S3DIS-H dataset as an example and report the efficiency metrics. In Table 10, the results (using PointNet++)

*Table 9.* Detailed hierarchy structures.

| Coarse Category (Parent) | Mapped Fine-Grained Categories (Children) |
|---|---|
| **S3DIS-H** | |
| (A) Static Elements | wall, floor, ceiling |
| (B) Openings | window, door |
| (C) Furniture | table, chair, sofa, bookcase |
| (D) Miscellaneous | beam, column, board, clutter |
| **SensatUrban-H** | |
| (A) Core Traffic Infrastructure | road, footpath, parking, bridge, rail |
| (B) Natural & Ground | vegetation, ground, water |
| (C) Traffic Elements | car, bike, unclassified |
| (D) Urban Amenities | building, street furniture, wall |

**Campus3D** (The original L1/L3/L5 hierarchy levels are re-indexed as L0/L1/L2 in the main text.)

*Level 1 → Level 3 Mapping:*

| L1: ground | L3: natural, play_field, path & stair, driving_road, man_made |
|---|---|
| L1: construction | L3: construction |
| L1: unclassified | L3: unclassified |

*Level 3 → Level 5 Mapping:*

| L3: natural | L5: natural |
|---|---|
| L3: play_field | L5: play_field, sheltered, un-sheltered, bus_stop |
| L3: path & stair | L5: path & stair |
| L3: driving_road | L5: car, bus, not_vehicle |
| L3: man_made | L5: man_made |
| L3: construction | L5: roof, wall, link, artificial landscape, lamp, others |
| L3: unclassified | L5: miscellaneous |

*Table 10.* Efficiency comparison on S3DIS-H dataset using PointNet++ backbone. Latency is measured by ms/sample, and throughput is samples/s.

| Method | Param. (M) | FLOPs (G) | Latency | Train Throughput | Infer Throughput |
|---|---|---|---|---|---|
| MTHS | 1.69 | 15.09 | 109.06 | 5.75 | 9.17 |
| DHL | 1.04 | 14.00 | 102.21 | 5.33 | 9.78 |
| ML3DHS (ours) | 1.72 | 15.50 | 109.22 | 3.60 | 9.16 |

indicate that the computational cost for inference of our method ML3DHS is at the same level as comparison methods. This is because the additional contrastive discrimination branches in our ML3DHS are used for training, and they will not be used for model inference. Therefore, the accuracy improvement is worth the efficiency cost.

### C.2. ANALYSIS OF MULTI-HIERARCHY CONFLICT

To further verify our method to alleviate the multi-hierarchy conflict by the late-decoupled framework (LDF), we remove CFG, CDB modules and measure the dynamics at the 30-th epoch of model training on S3DIS-H dataset.

**Algorithm 1** : Training Algorithm for ML3DHS

---

1: **Input:** 3D point cloud dataset $\mathcal{X}$, hierarchical labels $\{\hat{\mathbf{Y}}^{(h)}\}_{h=1}^{H}$, learning rate $\eta$, batch size $|B|$, trade-off $\lambda$, EMA $\beta$.
2: Randomly initialize the network parameters
3: **while** not reaching the maximal epochs **do**
4:     Sample a mini-batch $B \subset \mathcal{X}$
5:     Extract the features through the encoder module
6:     *// 1. Process Main Branch*
7:     **for** $h = 1 \rightarrow H$ **do**
8:         Compute main branch logits $\mathbf{Y}^{(h)}$.
9:         Apply coarse-to-fine guidance from the $(h-1)$-th hierarchy if $h > 1$.
10:     **end for**
11:     Compute main branch loss $\mathcal{L}_{\text{Ld-3DHS}}$ using cross-entropy ($\mathcal{L}_{\text{ces}}$) and consistency ($\mathcal{L}_{\text{chc}}$) objectives.
12:     *// 2. Process Auxiliary Branch*
13:     **for** each hierarchy $h$ **do**
14:         Extract class-specific features $\mathbf{F}^{(h,c)}$.
15:     **end for**
16:     With no gradient, update all main and auxiliary cluster prototypes via EMA.
17:     Compute auxiliary branch loss $\mathcal{L}_{\text{aux}}$ using contrastive loss ($\mathcal{L}_{\text{con}}$) and bi-branch self-supervised loss ($\mathcal{L}_{\text{bis}}$) over all hierarchies.
18:     *// 3. Joint Optimization and Cluster Updates*
19:     Compute the total loss $\mathcal{L}_{\text{total}} \leftarrow \mathcal{L}_{\text{Ld-3DHS}} + \lambda \mathcal{L}_{\text{aux}}$.
20:     Update network encoder parameters $\theta$ and late-decoupled individual decoder parameters $\{\delta^{(h)}\}_{h=1}^{H}$ via backpropagation on $\mathcal{L}_{\text{total}}$.
21: **end while**
22: **Output:** The main branch model (Ld-3DHS branch) for testing on the test set.

---

*Table 11.* Gradient dynamics analysis at the 30-th training epoch on S3DIS-H. GC(L0,L1): Gradient cosine between L0 and L1, where the gradient we measured on L0/1 is $\nabla_\theta \mathcal{L}_{ce}^{0/1}(\theta)$. GN(L0), GN(L1): Gradient norm mean of L0, L1. R(L0<L1): The ratio of samples whose gradient norm is greater at L1 than at L0.

| Variant | GC(L0,L1) | GN(L0), GN(L1) | R(L0<L1) | mIoU (%) on L0, L1 | Loss on L0, L1 |
|---|---|---|---|---|---|
| w/o LDF | 0.852 | 0.237, 0.348 | 0.980 | 52.0, 38.7 | 0.564, 0.880 |
| w/ LDF | 0.629 | 0.219, 0.244 | 0.726 | 52.9, 39.5 | 0.528, 0.820 |

The results in Table 11 demonstrate that LDF reduces the gradient cosine similarity between hierarchical losses (from 0.852 to 0.629), indicating less conflict between L0 and L1 (since GC close to 1 means gradients are strongly coupled, leading to mutual interference, while a lower GC means gradients are more independent and reduced their conflict). Moreover, the gradient norms become more balanced (from 0.237, 0.348 to 0.219, 0.244), and the ratio R(L0<L1) decreases significantly (from 0.980 to 0.726), suggesting that LDF alleviates the dominance of L1 gra-

*Table 12.* Per-class mIoU (%) on S3DIS-H. "#Points" denotes the total number of points ($\times 10^7$) for each class in the test set.

| Class (#Points) $\times 10^7$ | Floor (1.30) | Ceiling (1.54) | Clutter (0.71) | Window (0.28) | Door (0.24) | Column (0.14) |
|---|---|---|---|---|---|---|
| w/o CFG w/o CDB | 94.87 | 88.05 | 47.23 | 48.35 | 52.02 | 33.16 |
| w/ CFG w/o CDB | 95.04 | 89.61 | 50.58 | 51.09 | 54.96 | 36.38 |
| w/o CFG w/ CDB | 95.22 | 88.72 | 57.35 | 54.94 | 59.97 | 48.92 |
| **w/ CFG w/ CDB** | **95.13** | **90.28** | **63.46** | **61.27** | **63.56** | **52.47** |

*Table 13.* Performance comparison on ScanNet200 (PointNet++).

| Method | Avg. mIoU | $L_0$ (20 cls) | $L_1$ (40 cls) | $L_2$ (200 cls) |
|---|---|---|---|---|
| MTHS | 30.77 | 50.87 | 32.30 | 9.13 |
| DHL | 30.37 | 51.91 | 30.90 | 8.29 |
| **ML3DHS (ours)** | **32.93** | **54.17** | **34.32** | **10.31** |

dients on L0. This stabilization correlates with improved mIoU and reduced loss, providing direct evidence that LDF mitigates the multi-hierarchy conflict.

### C.3. MITIGATING CLASS IMBALANCE

To further verify our method to mitigate class imbalance issues by the coarse-to-fine guidance (CFG) and contrastive discrimination branch (CDB), we remove LDF module and then conduct the ablation experiments in Table 12 on S3DIS-H dataset (where floor, ceiling are majority classes and window, door, column are minority classes). The per-class mIoU results suggest that our CFG and CDB modules have significant promoting effect on segmentation especially for the minority classes.

### C.4. ADDITIONAL EVALUATION ON SCANNET200

We add the comparison experiments on ScanNet200 dataset (Dai et al., 2017). The dataset has three hierarchies where 200-40 class mapping follows the correspondence provided by the dataset itself (NYU40), and 40-20 class mapping is constructed by the method in DHL. We adopt the data preprocessing in Pointcept (Contributors, 2023) to make the data compatible with PointNet++ backbone. MTHS, DHL, and our ML3DHS use the above same settings, and the results in Table 13 indicate that our method consistently achieves better performance than competitors.

### C.5. CLARIFICATION ON CLUSTER AND CLUSTERING

We construct "semantic cluster prototypes" by the mean values of features $\mathbb{E}[\mathbf{h}_i^{(h,c)}]$ and $\mathbb{E}[\mathbf{f}_i^{(h,c)}]$. This is inspired by K-Means' approach to obtain clusters and thus we adopt the concept of "cluster" to describe our method. Constructing semantic cluster prototypes for contrastive learning is the self-supervised learning motivated part in our method. The auxiliary branch actually leverages the ground truth labels to compute the loss functions and thus there are differences between our method and unsupervised clustering.

