# OpenReview forum: "Multi-Label Learning with Contrastive Cluster Self-Supervision for 3D Hierarchical Semantic Segmentation"
_ICML.cc/2026/Conference — ICML 2026 regular_

### Official Review · Reviewer_WeMG · 2026-03-04

**Soundness:** 2
**Presentation:** 3
**Significance:** 3
**Originality:** 3
**Overall Recommendation:** 4
**Confidence:** 5

**Summary:**

This paper introduces ML3DHS to handle the multi-hierarchy conflict between learning multiple labels and a parameter-sharing model, and
the class imbalance issue hindering performance on minority classes.  First, the authors theoretically verified that multiple independent decoders outperform a shared decoder. Then, coarse-to-fine hierarchical guidance and cross-hierarchy consistency are used to improve the training of decoders at different hierarchy levels. 3DHS-oriented contrastive cluster self-supervision learning is proposed for class imbalance issue. Contrastive discrimination learns discriminative point cloud features for individual classes.  Smantic-cluster-based bi-branch
self-supervision mechanism promotes class-imbalance 3DHS tasks. ML3DHS achieves state-of-the-art 3DHS performance across multiple 3D benchmark datasets and backbones, and can be integrated into previous methods to further improve their 3DHS performance.

**Compliance With Llm Reviewing Policy:**

Affirmed.

**Final Justification:**

The rebuttal addresses most of my concerns, and I will maintain the positive rating.

**Key Questions For Authors:**

Is the mapping matrix A learnable?

**Limitations:**

The limitations section should be added.

**Strengths And Weaknesses:**

Strengths:
1 Although independent decoders have been applied in various scenarios, including 2D hierarchical semantic segmentation, none have theoretically verified their superiority over shared decoders. The verification provided here is a highlight.

2 The proposed improvement modules are reasonable and the ablation experiments are sufficient.

3 The effectiveness of the method has been validated on different backbones, indicating that this is a general design.

Weaknesses:

1 Section 2.2 introduces multiple methods for 3D hierarchical semantic segmentation, but only two methods are compared in Table 1.

2 No experiments were conducted on the ScanNet200 dataset.

3 Descriptions such as "the middle feature between Y(h-1) and Z" (200 and 201 lines) still make it hard to understand which part of the network this output corresponds to; it would be best to label them accordingly in Figure 1.

4 The limitations section should be added.

5 The introduction of the mapping matrix A is not detailed enough.

---

> ### Author Rebuttal · Authors · 2026-03-31
>
> We sincerely thank **Reviewer WeMG** for recognizing our work's value and for raising valuable questions. We responded to them one-by-one to address any potential concerns.
>
> >Q1: Section 2.2 introduces multiple methods for 3D hierarchical semantic segmentation, but only two methods are compared in Table 1.
>
> In Section 2.2, we reviewed the work related to 3D hierarchical semantic segmentation (3DHS), where the methods we mentioned (Farabet et al., 2013; Lin et al., 2017; Zhao et al., 2017) are hierarchical semantic segmentation approaches for 2D images and they cannot be directly applied to 3DHS tasks. At present, 3DHS methods are rare and we choose the representative MTHS and DHL for comparison. Meanwhile, the normal 3D segmentation methods (such as PointNet++, PointTFv2, PointTFv3) have also been adopted in our experiments, and they should be embedded into 3DHS frameworks to achieve hierarchical segmentation.
>
> >Q2: No experiments were conducted on the ScanNet200 dataset.
>
> In this response, we add the experiments on ScanNet200 dataset. The dataset has three hierarchies where 200-40 class mapping follows the correspondence provided by the dataset itself, and 40-20 class mapping is constructed by the method in DHL. We adopt the data preprocessing in ''Pointcept: A Codebase for Point Cloud Perception Research'' to make the data compatible with PointNet++ backbone. MTHS, DHL, and our ML3DHS use the above same settings, and the following results indicate that our method consistently achieves better performance than competitors.
>
> | | Avg.mIoU | L0 (20 classes) | L1 (40 classes)| L2 (200 classes)|
> | :--- | :---: | :---: | :---: | :---: |
> | MTHS          | 30.77 | 50.87 | 32.30 | 9.13  |
> | DHL           | 30.37 | 51.91 | 30.90 | 8.29  |
> | ML3DHS (ours) | **32.93** | **54.17** | **34.32** | **10.31** |
>
> >Q3: Descriptions such as "the middle feature between Y(h-1) and Z" (200 and 201 lines) still make it hard to understand which part of the network this output corresponds to; it would be best to label them accordingly in Figure 1.
>
> Thanks for this valuable suggestion and we will improve Figure 1 by adding feature descriptions in the network. Specifically, the features behind encoders are $\mathbf{Z}$, the middle features behind decoders are $\mathbf{H}^{h}$, the contrastive features behind MLP are $\mathbf{F}^{h,c}$, and the features behind classifier are $\mathbf{Y}^{h}$.
>
> >Q4: The limitations section should be added.
>
> We address our limitations in the conclusion section, specifically noting that the non-shared decoders and the extra auxiliary branch will increase the model size for training. The cost is affordable because the training part will not affect the inference effectiveness and our ML3DHS is proven to be useful in addressing the multi-hierarchy conflict as well as the class imbalance issue.
>
> >Q5: The introduction of the mapping matrix A is not detailed enough.
>
> The matrix $\mathbf{A}^{(h,h-1)}$ is fixed during model training and it represents the label mapping from the hierarchy $h$ to the hierarchy $(h-1)$. When class labels in the two hierarchies are definite, $\mathbf{A}^{(h,h-1)}$ derived using Eq.(6) is also determined.

---

> > ### Author Rebuttal · Reviewer_WeMG · 2026-04-07
> >
> > Thank you for the rebuttal. It addresses most of my concerns, and I will maintain the positive rating.

---

### Official Review · Reviewer_P9qw · 2026-03-10

**Soundness:** 3
**Presentation:** 3
**Significance:** 3
**Originality:** 3
**Overall Recommendation:** 5
**Confidence:** 4

**Summary:**

The paper introduces ML3DHS, a framework designed for 3D hierarchical semantic segmentation. It tackles two main problems: optimization conflicts that happen when a single shared model tries to learn multiple labels for each point, and the heavy class imbalance present in 3D scenes. The authors solve this by using a late-decoupled architecture, which shares an initial encoder but splits into separate decoders for each hierarchy, guided by a coarse-to-fine consistency mechanism. To fix the class imbalance, they add an auxiliary contrastive cluster self-supervision branch that groups similar points to provide extra training signals for rare classes. They evaluated the method on Campus3D, S3DIS-H, and SensatUrban-H across three backbones, achieving state-of-the-art results.

**Compliance With Llm Reviewing Policy:**

Affirmed.

**Final Justification:**

After reading the rebuttals by the authors I am satisfied with their work and all my concerns have been adequately answered, and I will maintain my score. I am looking forward to seeing the finalized paper with all the improvements made during the discussion period.

**Key Questions For Authors:**

Can you clarify the baseline number discrepancy for SensatUrban-H? In Table 1, the ML3DHS Avg. mIoU with PointNet++ is reported as 49.73%. However, in the ablation studies (Tables 5 and 6), the baseline variant (with all components active) lists the SensatUrban-H performance as 49.25%. Which number is correct, and what accounts for this difference?

While Figure 3 shows the training time cost , what is the exact impact on inference time and memory usage during deployment compared to the shared-decoder baselines? Clarifying this would help assess real-world applicability.

**Limitations:**

Yes. The authors adequately address their technical limitations in the conclusion, specifically noting that the non-shared decoders and the auxiliary branch increase the model size and training overhead. They also include a brief impact statement in the appendix, stating there are no specific negative societal consequences to highlight.

**Strengths And Weaknesses:**

Soundness: The technical approach is solid and well-supported. The inclusion of Theorem 1 gives a good theoretical backing for why the late-decoupled decoders resolve gradient conflicts compared to a shared decoder. The experiments are extensive, covering multiple datasets and backbones. However, there is a numeric discrepancy in the data reporting between the main results and the ablations.

Presentation: The paper is generally well-structured and easy to follow. The motivation is clear, and the figures help visualize the architecture and qualitative results.

Significance: 3DHS is highly relevant for embodied AI applications like robotics and autonomous driving. Showing that ML3DHS can be used as a plug-and-play module to improve existing baselines like MTHS and DHL shows its practical utility.

Originality: Applying a late-decoupled multi-label learning strategy combined with contrastive cluster self-supervision specifically to 3D hierarchical segmentation is a novel and effective combination of existing ideas.

---

> ### Author Rebuttal · Authors · 2026-03-31
>
> We sincerely thank **Reviewer P9qw** for recognizing our work's value. The following reply addresses the concerned numeric and inference efficiency questions.
>
> >Q1: There is a numeric discrepancy in the data reporting between the main results and the ablations. Can you clarify the baseline number discrepancy for SensatUrban-H? In Table 1, the ML3DHS Avg. mIoU with PointNet++ is reported as 49.73%. However, in the ablation studies (Tables 5 and 6), the baseline variant (with all components active) lists the SensatUrban-H performance as 49.25%. Which number is correct, and what accounts for this difference?
>
> We rechecked the experimental results and found that the reason why the results of SensatUrban-H in Tables 5,6 were inconsistent with those in Table 1 was that Tables 5,6 had not been updated timely due to different versions. The corrected results are as follows, which does not alter the conclusion that has already been reached in our paper.
>
> | LDF | CFG | CDB | SensatUrban-H | SensatUrban-H |
> |:---:|:---:|:---:|:--------------:|:--------------:|
> | ✓   | ✓   | ✓   | 49.25 (old)    | 49.73 (corrected)
> | ✗   | ✓   | ✓   | 47.74 (-1.51) | 47.74 (-1.99)
> | ✓   | ✗   | ✓   | 49.05 (-0.20) | 49.05 (-0.68)
> | ✓   | ✓   | ✗   | 48.15 (-1.10) | 48.15 (-1.58)
>
> | $\mathcal{L}_{con}$ | $\mathcal{L}_{chc}$ | $\mathcal{L}_{bis}$ | SensatUrban-H | SensatUrban-H |
> |:---:|:---:|:---:|:--------------:|:--------------:|
> | ✓   | ✓   | ✓   | 49.25 (old)   | 49.73 (corrected)
> | ✗   | ✓   | ✓   | 48.75 (-0.50) | 48.75 (-0.98)
> | ✓   | ✗   | ✓   | 47.50 (-1.75) | 47.50 (-2.23)
> | ✓   | ✓   | ✗   | 48.15 (-1.10) | 48.15 (-1.58)
>
> >Q2: While Figure 3 shows the training time cost , what is the exact impact on inference time and memory usage during deployment compared to the shared-decoder baselines? Clarifying this would help assess real-world applicability.
>
> To clarify this question, we report the inference parameter, FLOPs, and latency metrics on S3DIS-H dataset with PointNet++ as follows. The results indicate that our method ML3DHS's computational cost for inference is at the same level as the baseline methods with shared-decoder. This is mainly because the additional contrastive discrimination branches in our ML3DHS are used for training, and they will not be used for model inference.
>
> |  | parameter (M) | FLOPs (G) | latency (ms/sample) |
> | :--- | :---: | :---: | :---: |
> | MTHS          | 1.69 | 15.09 | 109.06 |
> | DHL           | 1.04 | 14.00 | 102.21 |
> | ML3DHS (ours) | 1.72 | 15.50 | 109.22 |

---

> > ### Author Rebuttal · Reviewer_P9qw · 2026-03-31
> >
> > Thank you for the straightforward rebuttal. Both of my questions have been fully resolved.
> >
> > Numeric Discrepancy: It is completely understandable that the mismatch was a version control error. Just please ensure that the SensatUrban-H baseline numbers in Tables 5 and 6 are updated to match the 49.73% reported in Table 1 for the camera-ready version.
> >
> > Inference Efficiency: Providing the exact parameters, FLOPs, and latency metrics directly addresses my concern about real-world applicability. Thank you for confirming that the auxiliary contrastive discrimination branch is dropped during deployment.

---

### Official Review · Reviewer_y33R · 2026-03-12

**Soundness:** 3
**Presentation:** 3
**Significance:** 3
**Originality:** 3
**Overall Recommendation:** 5
**Confidence:** 4

**Summary:**

The authors propose a novel multi-label learning with contrastive cluster self-supervision framework for 3D hierarchical semantic segmentation. Specifically, the authors contribute with a late-decoupled multi-label learning network, which employs multiple decoders with the coarse-to-fine hierarchical guidance and consistency.
Furthermore, they introduce a flexible cluster self-supervised strategy, aiming to learn cluster-wise point cloud features with contrastive loss. Experimental results demonstrate superior effectiveness with respect to competitive methods on benchmark datasets.

**Compliance With Llm Reviewing Policy:**

Affirmed.

**Final Justification:**

the rebuttal addressed my main concerns, so I upgraded my final decision

**Key Questions For Authors:**

The authors state that their method could be useful for domain-specific applications, including autonomous driving. However, in such scenarios, achieving strong efficiency in addition to effectiveness is often crucial. In this regard, a key question concerns how the proposed model performs in terms of computational efficiency and inference latency, particularly in real-time or time-critical settings.

**Limitations:**

Efficiency analysis. The authors limit their efficiency analysis to the training stage. However, given that the target application scenarios may include time-critical environments, it would be important to evaluate the inference efficiency of the proposed framework. Providing measurements such as inference time or throughput would help better assess the practical applicability of the method.

**Strengths And Weaknesses:**

Strengths:
The paper is well structured and well motivated. The authors demonstrate how to intervene to solve critical issues relevant to the field, suggesting also the potential applicability of their proposed method (e.g., autonomous driving). The authors have adequately assessed both quantitative evaluation and benchmarked their method against other cutting-edge frameworks on different datasets. Furthermore, the authors have also provided quantitative ablation studies, confirming the effectiveness of the proposed modules.

Weaknesses:
A first minor criticism regards the presentation of the related work section. Given its fragmented organization, it is a little bit hard to follow both the evolution of relevant methodologies and how effectively the authors have proposed to solve the underlying research drawbacks. The manuscript contains some minor theoretical errors. Specifically, delving into the methodology section (Sec. 3), Y is unlikely to be used to represent predictions, as it is usually referenced as \cap{Y}. Instead, within this work, it is rather the ground truth tensor
to be referenced as \cap{Y} . Furthermore, some formulations are quite unclear.
For example, the construction of the incidence matrix A is unclear, since there is no true equation determining it (e.g., A = · · ·). Some terminology is not perfectly aligned. The authors say that they propose a cluster self-supervised module. However, they leverage the actual GTs to compute L_con and L_bis, respectively. This is misleading, as cluster-based indicates a label-free approach.
The proposed method is rather prototype-based.

---

> ### Author Rebuttal · Authors · 2026-03-31
>
> We sincerely thank **Reviewer y33R** for recognizing our work's value and for giving such meticulous comments. We responded to all of them as follows.
>
> >Q1: A first minor criticism regards the presentation of the related work section. Given its fragmented organization, it is a little bit hard to follow both the evolution of relevant methodologies and how effectively the authors have proposed to solve the underlying research drawbacks.
>
> We will improve the related work section to better show the methodology evolution, e.g., from traditional 3D segmentation methods to recent 3D hierarchical segmentation methods. Existing methods still face the challenges of balancing multiple hierarchies and classes, which motivates us to develop our new method ML3DHS.
>
> >Q2: Delving into the methodology section (Sec. 3), Y is unlikely to be used to represent predictions, as it is usually referenced as \cap{Y}. Instead, within this work, it is rather the ground truth tensor to be referenced as \cap{Y}.
>
> We will modify the notations to make the paper conform to the standard expression format.
>
> >Q3: The construction of the incidence matrix A is unclear, since there is no true equation determining it (e.g., A = · · ·).
>
> Since class labels in each hierarchy are definite, the label mapping between two hierarchies is also definite. To serve our proposed cross-hierarchy consistency objective in Eq.(7), we denote $\mathbf{A}^{(h,h-1)}$ as the mapping matrix. Based on Eq.(6), its closed-form solution can be expressed as $\mathbf{A}^{(h,h-1)} = \left( \hat{\mathbf{Y}}^{(h-1)\dagger} \, \hat{\mathbf{Y}}^{(h)} \right)^T$, where $\dagger$ denotes pseudoinverse.
>
> >Q4: Some terminology is not perfectly aligned. The authors say that they propose a cluster self-supervised module. However, they leverage the actual GTs to compute L_con and L_bis, respectively. This is misleading, as cluster-based indicates a label-free approach. The proposed method is rather prototype-based.
>
> As illustrated in Lines 239-248, we construct ''semantic cluster prototypes'' by the mean values of features $\mathbb{E}[\mathbf{h}_i^{(h,c)}]$ and $\mathbb{E}[\mathbf{f}_i^{(h,c)}]$. This is inspired by K-Means' approach to obtain clusters and thus we adopt the concept of ''cluster'' to describe our method. We will improve the terminology for clarifying the difference between our method and unsupervised clustering, and thanks for this comment again.
>
> >Q5: The authors limit their efficiency analysis to the training stage. However, given that the target application scenarios may include time-critical environments, it would be important to evaluate the inference efficiency of the proposed framework. Providing measurements such as inference time or throughput would help better assess the practical applicability of the method.
>
> To address this concern, we additionally report more efficiency metrics using PointNet++ point cloud backbone. For example, on S3DIS-H dataset, the following comparison indicates that our method ML3DHS's computational cost for inference is at the same level as comparison methods. For a point cloud scene, the inference latency is about ~100 ms and thus it achieves real-time efficiency.
>
> |  | inference parameter (M) | FLOPs (G) | inference latency (ms/sample) | training throughput (samples/s) | inference throughput (samples/s) |
> | :--- | :---: | :---: | :---: | :---: | :---: |
> | MTHS          | 1.69   | 15.09 | 109.06 | 5.75 | 9.17 |
> | DHL           | 1.04   | 14.00 | 102.21 | 5.33  | 9.78 |
> | ML3DHS (ours) | 1.72   | 15.50 | 109.22 | 3.60  | 9.16 |

---

> > ### Author Rebuttal · Reviewer_y33R · 2026-04-07
> >
> > The authors have addressed my concerns.

---

### Official Review · Reviewer_QJ1b · 2026-03-13

**Soundness:** 3
**Presentation:** 3
**Significance:** 3
**Originality:** 3
**Overall Recommendation:** 4
**Confidence:** 3

**Summary:**

This paper studies 3D Hierarchical Semantic Segmentation (3DHS), aiming to predict multi-level semantic labels for each point in the point cloud, ranging from coarse to fine. The author believes that existing methods mainly face two core issues: multi-level shared parameters can lead to optimization conflicts, while the prevalent class imbalance across different levels can bias the model towards the majority class.
To address these issues, the paper proposes ML3DHS, which adopts a structure of "shared encoder + late-decoupled decoder" and incorporates coarse-to-fine hierarchical guidance and cross-hierarchy consistency to reduce learning conflicts between different levels.
In addition, the author has designed a contrastive clustering self-supervised branch for 3DHS, which enhances the discriminative ability for minority classes and fine-grained categories by contrastively learning intra-class features, constructing semantic prototypes, and performing dual-branch self-supervision.

**Compliance With Llm Reviewing Policy:**

Affirmed.

**Final Justification:**

After reading the rebuttal, I am satisfied that my concerns have been addressed, and I therefore maintain my positive score. That said, I believe the paper would benefit from incorporating the main clarifications and supporting details from the rebuttal into the final manuscript, which would further enhance its clarity, completeness, and overall quality.

**Key Questions For Authors:**

The current experiment seems more like it is demonstrating that "performance is better with the addition of these modules," rather than fully explaining "why they can alleviate multi-hierarchy conflict" or "why they can specifically improve minority classes." If the paper could include more direct supporting evidence, it would be significantly more convincing, such as: gradient conflict metrics between different levels, a comparison of optimization dynamics between shared decoders and decoupled decoders during training, more granular statistics for long-tail/rare-class, performance change curves based on the frequency of different categories, or more intuitive feature visualization and failure case analysis.
Therefore, I believe that the main deficiency of this paper lies not in the weak results themselves, but in the insufficiently solid argumentation at the mechanism level: it has demonstrated the effectiveness of the method, but it has not yet convincingly explained to what extent the two core issues proposed have been addressed, and whether this solution possesses generalizable explanatory power.

**Limitations:**

yes

**Strengths And Weaknesses:**

Strengths

This paper focuses on 3D hierarchical semantic segmentation (3DHS), a problem that inherently carries significant importance and practical relevance. The author explicitly outlines two core challenges: firstly, multi-hierarchy conflict is prone to occur in parameter-shared models with multi-level labels, and secondly, category imbalance in multi-level scenarios can compromise the performance of minority classes. The entire work revolves around these two issues, with a relatively clear definition of the problems.

The method design is relatively comprehensive and intuitive overall. The paper proposes a shared encoder, late-decoupled decoders, coarse-to-fine guidance, and an additional contrastive cluster self-supervision branch. The division of labor among these modules is relatively clear: the former is mainly used to alleviate optimization conflicts between different levels, while the latter is mainly used to enhance discriminability in class-imbalanced scenarios.

The paper provides certain theoretical and experimental support. For instance, it presents a theoretical argumentation regarding the superiority of the late-decoupled decoder over the shared decoder, and illustrates through structural ablation and loss ablation that LDF, CFG, CDB, Lcon, Lchc, and Lbis all contribute to the final performance, which makes the method more than just an empirical stacking of modules.

Weaknesses

The analysis of the computational cost brought by the additional branches in the paper is still insufficient. The author himself acknowledges that non-shared decoders and extra auxiliary branches increase the model size during training; the main text only qualitatively states that its time consumption is "slightly higher than" the baseline, attributing it to the additional contrastive discrimination branch. However, the paper does not further report more complete efficiency metrics such as parameter count, FLOPs, memory usage, training throughput, inference latency, or inference-time throughput. Therefore, it is currently difficult to comprehensively judge whether this accuracy improvement is worth the corresponding efficiency cost.

Furthermore, although the paper positions its core contribution as addressing "the multi-hierarchy conflict between learning multiple labels and a parameter-sharing model" and "the class imbalance issue hindering performance on minority classes", the evidence provided so far remains indirect. For the former, the authors primarily rely on theoretical statements, late-decoupled design, and the performance degradation after removing LDF in ablation studies to support their claim; for the latter, they mainly rely on the design motivation of the contrastive branch, the performance degradation after removing CDB/Lcon/Lbis, and a small number of per-class results. Such evidence suggests that these modules are "effective", but it is not sufficient to unequivocally prove that they indeed directly address the two claimed issues separately.

---

> ### Author Rebuttal · Authors · 2026-03-31
>
> We sincerely thank **Reviewer QJ1b** for recognizing our work's value and for raising valuable questions. We responded to them one-by-one to address any potential concerns.
>
> >Q1: The paper does not report more complete efficiency metrics such as parameter count, FLOPs, memory usage, training throughput, inference latency, or inference-time throughput.
>
> To report these efficiency metrics, we take S3DIS-H dataset as an example and report the following table. The results (using PointNet++) indicate that the computational cost for inference of our method ML3DHS is at the same level as comparison methods. This is because the additional contrastive discrimination branches in our ML3DHS are used for training, and they will not be used for model inference. Therefore, the accuracy improvement is worth the efficiency cost.
>
> | | inference parameter (M) | FLOPs (G) | inference latency (ms/sample) | training throughput (samples/s) | inference throughput (samples/s) |
> | :--- | :---: | :---: | :---: | :---: | :---: |
> | MTHS | 1.69 | 15.09 | 109.06 | 5.75 | 9.17 |
> | DHL | 1.04 | 14.00 | 102.21 | 5.33 | 9.78 |
> | ML3DHS (ours) | 1.72 | 15.50 | 109.22 | 3.60 | 9.16 |
>
> > Q2: Although the paper positions its core contribution as addressing "the multi-hierarchy conflict between learning multiple labels and a parameter-sharing model" and "the class imbalance issue hindering performance on minority classes", the evidence provided so far remains indirect. For the former, the authors primarily rely on theoretical statements, late-decoupled design, and the performance degradation after removing LDF in ablation studies to support their claim; for the latter, they mainly rely on the design motivation of the contrastive branch, the performance degradation after removing CDB/Lcon/Lbis, and a small number of per-class results. Such evidence suggests that these modules are "effective", but it is not sufficient to unequivocally prove that they indeed directly address the two claimed issues separately.
>
> To further verify our method to alleviate the multi-hierarchy conflict by the late-decoupled framework (LDF), we remove CFG, CDB modules and measure the dynamics at the 30-th epoch of model training on S3DIS-H dataset as follows.
>
> GC(L0,L1): Gradient cosine between L0 and L1, where the gradient we measured on L0/1 is $\nabla_{\theta}\mathcal{L}^{0 / 1}_{ces}(\theta)$.
>
> GN(L0), GN(L1): Gradient norm mean of L0, L1.
>
> R(L0<L1): The ratio of samples whose gradient norm is greater at L1 than at L0.
>
> | At the 30-th epoch| &emsp; GC(L0,L1)  | &emsp; GN(L0),GN(L1)  | &emsp; R(L0<L1)  | &emsp; mIoU(%) on L0,L1  | &emsp; Loss on L0,L1 |
> |---|---:|---:|---:|---:|---:|
> | w/o LDF    |  0.852 | 0.237, 0.348 | 0.980 | 52.0, 38.7 | 0.564, 0.880 |
> | w/ LDF     |  0.629 | 0.219, 0.244 | 0.726 | 52.9, 39.5 | 0.528, 0.820 |
>
> These results demonstrate that LDF reduces the gradient cosine similarity between hierarchical losses (from 0.852 to 0.629), indicating less conflict between L0 and L1 (since GC close to 1 means gradients are strongly coupled, leading to mutual interference, while a lower GC means gradients are more independent and reduced their conflict). Moreover, the gradient norms become more balanced (from 0.237, 0.348 to 0.219, 0.244), and the ratio R(L0<L1) decreases significantly (from 0.980 to 0.726), suggesting that LDF alleviates the dominance of L1 gradients on L0. This stabilization correlates with improved mIoU and reduced loss, providing direct evidence that LDF mitigates the multi-hierarchy conflict.
>
> To further verify our method to mitigate class imbalance issues by the coarse-to-fineguidance (CFG) and contrastive discrimination branch (CDB), we remove LDF module and then conduct the following ablation experiments on S3DIS-H dataset (where floor, ceiling are majority classes and window, door, column are minority classes). The per-class mIoU results suggest that our CFG and CDB modules have significant promoting effect on segmentation especially for the minority classes. The ''floor (1.30×10⁷)'' denotes that the floor class has total ~1.30×10⁷ points in the test set.
>
> | | floor (1.30×10⁷) | ceiling (1.54×10⁷) | clutter (0.71×10⁷) | window (0.28×10⁷) | door (0.24×10⁷) | column (0.14×10⁷) |
> |---------|--------------|----------------|----------------|---------------|-------------|---------------|
> | w/  CFG w/  CDB  | **95.13** | **90.28** | **63.46** | **61.27** | **63.56** | **52.47** |
> | w/  CFG w/o CDB  | 95.04 | 89.61 | 50.58 | 51.09 | 54.96 | 36.38 |
> | w/o CFG w/  CDB  | 95.22 | 88.72 | 57.35 | 54.94 | 59.97 | 48.92 |
> | w/o CFG w/o CDB  | 94.87 | 88.05 | 47.23 | 48.35 | 52.02 | 33.16 |

---

> > ### Author Rebuttal · Reviewer_QJ1b · 2026-04-03
> >
> > Thanks for your response. It addresses most of my concerns, and I will maintain my positive rating.

---

### Decision · Program_Chairs · 2026-04-30

**Decision:**

Accept (regular)

**Comment:**

In this paper the authors presented a framework for 3D hierarchical semantic segmentation, with multi-label learning and contrastive cluster self-supervision. The paper was reviewed by four expert reviewers, followed by a rebuttal and discussion between the reviewers and authors. The paper received an overall positive rating with 2 Accept and 2 Weak Accept.

The reviewers agree on the motivation, the potential practical value, the technical design, and sufficient empirical evidence support. Good contributions could be made to the community, as appreciated by the reviewers.
There were some concerns regarding the weaknesses of the paper, including the computational cost, lack of sufficient convincing evidence, missing experimental comparisons and analysis, and some minor presentation issues. After the rebuttal and the author-reviewer discussion phase, most concerns were well addressed, as acknowledged by the reviewers.
Overall, the paper is technically sound, well-written, and could contribute to and be of interest to at least some fraction of the ICML community.

As a result, the AC is happy to recommend acceptance of this paper, but the authors are asked to incorporate the revisions agreed and the necessary additional justifications in the rebuttal/discussion to their final version.